# Scutellarein Induces Fas-Mediated Extrinsic Apoptosis and G2/M Cell Cycle Arrest in Hep3B Hepatocellular Carcinoma Cells

**DOI:** 10.3390/nu11020263

**Published:** 2019-01-24

**Authors:** Sang Eun Ha, Seong Min Kim, Ho Jeong Lee, Preethi Vetrivel, Venu Venkatarame Gowda Saralamma, Jeong Doo Heo, Eun Hee Kim, Sang Joon Lee, Gon Sup Kim

**Affiliations:** 1Research Institute of Life Science and College of Veterinary Medicine, Gyeongsang National University, Jinju 52828, Korea; sangdis2@naver.com (S.E.H.); ksm4234@naver.com (S.M.K.); preethivetrivel05@gmail.com (P.V.); gowdavenu27@gmail.com (V.V.G.S.); 2Gyeongnam Department of Environment Toxicology and Chemistry, Biological Resources Research Group, Korea Institute of Toxicology, 17 Jegok-gil, Jinju 52834, Korea; hojeong.lee@kitox.re.kr (H.J.L.); jdher@kitox.re.kr (J.D.H.); 3Department of Nursing Science, International University of Korea, Jinju 52833, Korea; iuknurse@nate.com

**Keywords:** scutellarein, flavonoid, hepatocellular carcinoma, apoptosis, cell cycle arrest

## Abstract

Scutellarein (SCU), a flavone found in the perennial herb *Scutellaria baicalensis*, is known for a wide range of biological activities. In the present study, we investigated the effects of treatment with SCU flavonoids on inducing apoptosis via the extrinsic pathway in Hep3B cells. SCU treatment significantly inhibited Hep3B cell proliferation and induced G2/M phase cell cycle arrest by inhibiting the expression level of the proteins Cdc25C, cdk1 and Cyclin B1. Allophycocyanin (APC)/Annexin V and propidium iodide (PI) double-staining showed upregulation of apoptotic cell death fraction. We further confirmed apoptosis by 4′-6-diamidino-2-phenylindole (DAPI) fluorescent staining and observed DNA fragmentation with agarose gel electrophoresis. Further, immunoblotting results showed that treatment with SCU showed no changes in Bax and Bcl-xL protein levels. In addition, SCU treatment did not affect the mitochondrial membrane potential (MMP) in Hep3B cells. On the contrary, treatment with SCU increased the expression of Fas and Fas ligand (FasL), which activated cleaved caspase-8, caspase-3, and polymeric adenosine diphosphate ribose (PARP), whereas the expression level of death receptor 4 (DR4) decreased. We confirmed that the proteins expressed upon treatment with SCU were involved in the Fas-mediated pathway of apoptosis in Hep3B cells. Thus, our findings in the current study strongly imply that SCU can be a basic natural source for developing potent anti-cancer agents for hepatocellular carcinoma (HCC) treatment.

## 1. Introduction

Hepatocellular carcinoma (HCC) is the most common form of primary liver cancer in adults, and it has high malignancy and poor prognosis [1]. HCC treatment options and prognosis depend on many factors such as tumor size and stage of tumor severity. Surgical treatment for HCC does not give significant outcomes because the cancer can be removed through surgery in only 10%–20% of patients [2]. Chemotherapy is one of the options that has been used to cure HCC. Thus, the current HCC research focus is on identifying alternative therapeutic drugs with high pharmacological effects and low normal cell toxicity [3].

Flavonoids are natural polyphenolic compounds that are known for their significant biological beneficial properties (including anti-cancer, anti-allergic, and anti-inflammatory activities), and it is also reported that they have low toxicity. The anti-cancer activities of flavonoids relate to their role in cell cycle arrest and apoptosis, preventing cell invasion and adhesion. The safety and efficacy of flavonoids as anti-cancer agents have been identified in human clinical studies [4]. *Scutellaria baicalensis* is one of the fundamental herbs used in traditional Chinese medicine, and it has a wide range of biological activities, such as anti-inflammation and anti-diarrheal effects [5,6]. The leaves and flowers of *Scutellaria baicalensis* contain flavonoid glycosides such as scutellarein (5,6,7,4′-tetrahydroxyflavone; SCU), a flavone found in the perennial herb that has a range of biological activities. Previous studies demonstrated that the whole flavonoid extract of SCU along with the monomer compound, have a broad scale of biological activities such as anti-oxidant, anti-inflammatory, and anticancer effects through the induction of apoptosis [7].

Cell cycle arrest and induction of apoptosis are important strategies in cancer therapeutic strategies [8]. The ability to modulate life or death of a cell is known for therapeutic potential in treating cancer cells. Thus, the focus of research has been directed toward the cell cycle and programmed cell death mechanisms [9]. The regulating factors of cell cycle processes are frequently modified in human cancer cells. The cyclin-dependent kinases (CDKs) are central players that control the initiation, progression, and completion of the cell cycle. Inhibiting CDK activity is expected to obstruct cell cycle events and lead to cell cycle arrest. Many compounds operate as anti-cancer agents at multiple steps in the cell cycle [10].

Apoptosis is generally defined as programmed cell death, and it plays important roles in developing and maintaining tissue homeostasis and cancer chemoprevention. Apoptosis is characterized by several distinct morphological features such as cell membrane blebbing, cell shrinkage, chromatin condensation, and DNA fragmentation, followed by the engulfment of macrophages [11].

The mechanism of apoptosis follows two distinct pathways: the extrinsic death receptor-mediated pathway and the intrinsic mitochondria-mediated pathway. Caspases are the central effectors of apoptosis and the two pathways that lead to other proteases and nucleases to cause apoptosis [12]. In the extrinsic apoptosis pathway, the Fas ligand (FasL) is upregulated when the cell-surface death receptor, Fas, is activated. The activation of the Fas leads to sequential activation of caspase-8, caspase-3, and polymeric adenosine diphosphate ribose (PARP). In the intrinsic apoptosis pathway, the release of diverse apoptotic stimuli from intrinsic signals including those from DNA damage and oxidative stress converge to the mitochondria and then lead to the release of cytochrome c from the mitochondria to cytoplasm, initiating the caspase cascades [13].

In this study, we identified the anti-cancer effect of SCU in human hepatoma Hep3B cells. We found evidence that SCU prevented cell proliferation via cell cycle arrest in the G2/M phase and induction of the extrinsic apoptosis pathway in Hep3B cells. These findings suggest that SCU can be used for developing potent anti-cancer agents for HCC treatment.

## 2. Materials and Methods

### 2.1. Chemicals and Reagents

3-(4,5-Dimethylthiazol-2-yl)-2,5-diphenyltetrazolium bromide (MTT) was obtained from Duchefa Biochemie (Haarlem, the Netherlands). Antibodies to caspase-3, -8, and -9, cleaved caspase-3, -8, and -9, polymeric adenosine diphosphate ribose (PARP), cleaved PARP, Fas, FasL, Cyclin B1, Cdc25C, and Bcl-xL were purchased from Cell Signaling Technology (Danvers, MA, USA). Death receptor 4 (DR4) antibodies were obtained from Santa Cruz Biotechnology (Santa Cruz, CA, USA). Antibodies cdk1, Bax, and β-actin were purchased from Millipore (Temecula, CA, USA). 

### 2.2. Cell Culture and Scutellarein (SCU) Treatment

Human hepatocarcinoma cell line Hep3B was obtained from the Korea Cell Line Bank (Seoul, Korea). Dulbecco’s modified Eagle’s medium (DMEM), fetal bovine serum (FBS), phosphate-buffered saline (PBS), and antibiotics penicillin/streptomycin (P/S) were purchased from Gibco (BRL Life Technologies, Grand Island, NY, USA). Mycoplasma free Hep3B cells were cultured in DMEM supplemented with 10% FBS and 1% P/S at 37 °C in a humidified atmosphere of 5% CO_2_. To confirm mycoplasma contamination, we used the e-Myco™ Mycoplasma PCR Detection kit (iNtRON Biotechnology, Seoul, Korea). We cultured Hep3B cells for no more than 15 passages or 2 months. Scutellarein (SCU) was purchased from Chengdu Biopurify Phytochemicals Ltd. (Chengdu, Sichuan, China). Cells grown to 80% confluence were untreated (DMSO) or treated with indicated concentration of SCU for 24 h in complete media.

### 2.3. Cell Viability Assay

Cell viability was measured using MTT assay. Cells were seeded at 5 × 10^4^ cells in a 48-well plate and incubated overnight, followed by treatment with SCU at the concentrations of 0-, 100-, 200-, 300-, 400-, 500-, and 600-μM for 24 h. After incubation, 50 μL of MTT (0.5 mg/mL) solution was added to each well and incubated for about 3 h at 37 °C. The formazan precipitate formed after incubation was dissolved in 300 μL of DMSO and the absorbance of converted dye was measured at a wavelength of 540 nm with a micro-plate reader (BioTek, Winooski, VT, USA). Cell viability was expressed as a percentage of proliferation versus SCU untreated group.

### 2.4. Morphological Change and DAPI Fluorescent Staining

For nuclear morphological analysis, Hep3B cells were plated on 12-well plates at 1 × 10^5^ cells after treatment with various concentrations of SCU (0-, 100-, 200-, and 300-μM) at 37 °C for 24 h. The cells were washed with ice-cold PBS and then fixed with 37% formaldehyde (1:4 dilutions with PBS) for 15 min at room temperature. Subsequently, the fixed cells were washed with PBS and stained with a 4′,6-diamidino-2-phenylindole (DAPI; Vectashield H-1500; Vector Laboratories, Burlingame, CA, USA). The nuclear morphology of the cells was examined by fluorescence microscopy (EVOS^®^, Life Technologies, Darmstadt, Germany).

### 2.5. DNA Fragmentation Assay

Hep3B cells were plated on 60-mm plates at 4 × 10^5^ cells after treatment with indicated concentrations of SCU (0-, 100-, 200-, and 300-μM) for 24 h and DNA was isolated from the cells harvested and lysed using lysis buffer containing 1% NP-40, 20 mM ethylenediaminetetraacetate (EDTA), 50 mM Tris-HCl, and pH 7.5 for 30min. The lysed cells were centrifuged at 3,000 rpm for 5 min and the supernatant was collected. The collected supernatant was incubated with 10 μL of 10% sodium dodecyl sulfate (SDS) solution and 5 μL of 100 mg/mL RNase A for 2 h at 56 °C in a water bath. Further, protein digestion was carried out by adding 10 μL of 25 mg/mL proteinase K enzyme and incubated for 2 h at 37 °C, followed by the addition of 65 μL of 5 M NaCl and 500 μL of ice-cold ethanol, and the contents were mixed thoroughly. The mixture was further subjected to incubation for 2 h at −80 °C. It was then centrifuged for 20 min at 12,000 rpm followed by washing the pellet with 1 mL of 80% ice-cold ethanol and air-dried for 10 min at room temperature. The pellet was dissolved with 20 μL of Tris-EDTA (TE) buffer. The total DNA sample was then subjected to 1.5% agarose gel electrophoresis and DNA bands were visualized in UV light absorbance.

### 2.6. Annexin V Propidium Iodide Apoptosis Detection

Apoptotic cells were detected by using allophycocyanin (APC)/Annexin V apoptosis detection kit according to the manufacturer’s protocol (BD Biosciences, San Diego, CA, USA). Briefly, cells were plated on 60-mm plates at 4 × 10^5^ cells and then incubated with various concentrations of SCU (0-, 100-, 200-, and 300-μM), for 24 h. The cells were collected and washed with PBS re-suspended in binding buffer. The cells were stained with APC/Annexin V and propidium iodide (PI) for 15 min at room temperature in the dark, prior to the addition of binding buffer. Flow cytometry analysis was performed on the cell suspensions and the data obtained were analyzed using a fluorescence-activated cell sorting machine (FACSVerse^TM^ flow cytometer; BD Biosciences, Franklin Lakes, New Jersey, USA). In total, 10,000 events per sample were sorted and the data were analyzed using BD FACSuite^TM^ software (BD Biosciences, Becton & Dickson, Mountain View, CA, USA).

### 2.7. Analysis of Cell Cycle Distribution or DNA Content by Flow Cytometry

Flow cytometry was performed to analyze the cell cycle distribution and cell death. Hep3B cells were treated with various concentrations of SCU (0-, 100-, 200- and 300-μM) for 24 h at 37 °C. The cells were washed with ice-cold PBS. After incubation, trypsinized cells were collected in a 15-mL conical tube, and the pellet was obtained by centrifugation at 1200 rpm for 4 min. The pellets were washed twice with ice-cold PBS and fixed in 70% ice-cold ethanol for 1 h at −20 °C. The cell suspension was centrifuged further and the cells were washed in PBS and re-suspended in 400 μL of PBS containing 50 μg/mL PI (Sigma-Aldrich, St. Louis, MO, USA) and 50 μg/mL RNase A, followed by incubation in dark conditions for 15 min at room temperature. After incubation, flow cytometry analysis was performed on the cell suspensions and the data obtained were analyzed using FACSVerse^TM^ flow cytometer (BD Biosciences, Franklin Lakes, NJ, USA). The acquired FACS data was analyzed using ModFit LT software (Verity Software House, Topsham, ME, USA). 

### 2.8. Analysis of Mitochondrial Membrane Potential Using JC-1 Staining

Flow cytometry analysis was performed to detect the mitochondrial membrane potential in Hep3B cells. The cells were treated with various concentrations of SCU (0-, 100-, 200- and 300-μM) for 24 h at 37 °C. After treatment, the cells were stained with 10 μM the fluorescent dye 5,5′,6,6′-Tetrachloro-1,1′,3,3′-tetraethylbenzimidazolocarbocyanine iodide (JC-1 dye; R&D Systems, Minneapolis, MN, USA) and incubated for about 10 min at 37 °C. The cells were harvested by centrifucation at 1200 rpm for 5 min. The cell pellet obtained was suspended in serum free DMEM media. Flow cytometry analysis was performed on the cell suspensions using FACSVerseTM flow cytometer (BD Biosciences, Franklin Lakes, NJ, USA). The acquired FACS data was analyzed using BD FACSuiteTM software (BD Biosciences, Becton & Dickson, Mountain View, CA, USA).

### 2.9. Western Blotting Analysis

Hep3B cells were seeded in 60-mm plates at 4 × 10^5^ per well and treated with indicated concentrations of SCU (0-, 100-, 200-, and 300-μM) for 24 h at 37 °C. Cells were lysed using radioimmuno-precipitation assay (RIPA) buffer (iNtRON Biotechnology, Seoul, Korea) containing phosphatase and protease inhibitor cocktail (Thermo Scientific, Rockford, IL, USA) for 30 min in ice. Protein lysates were centrifuged at 10,000 rpm for 10 min at 4 °C and the protein concentrations were determined using a Pierce™ BCA assay (Thermo Fisher Scientific, Rockford, IL, USA). The protein extract was mixed with required volume of 5X sample buffer and kept at 100 °C for 5 min. Protein samples were resolved on 8%–15% SDS polyacrylamide gels and subjected to electrophoresis. The separated proteins were electro transferred to a polyvinylidene difluoride (PVDF) membrane. Membranes were blocked with 5% (bovine serum albumin (BSA) in Tris-buffered saline containing 1% Tween 20 (TBS-T, pH 7.4) at room temperature for 1 h, and incubated overnight at 4 °C with β-actin (1:10,000), Cdc25C (1:1000), cdk1 (1:1000), Cyclin B1 (1:1000), Bax (1:1000), Bcl-xL (1:1000), caspase-9 (1:1000), cleaved caspase-9 (1:500), DR4 (1:500), Fas (1:1000), FasL (1:500), caspase-8 (1:1000), cleaved caspase-8 (1:500), caspase-3 (1:1000), cleaved caspase-3 (1:500), and PARP (1:1000) primary antibody. The membranes were washed with TBS-T buffer for every 10 min in five repetitions at room temperature. Furthermore, they were incubated with 1:1000 dilution of horseradish peroxidase (HRP)-conjugated secondary antibody for 3 h at room temperature. The obtained protein blots were developed under an electrochemiluminescence (ECL) detection system (Bio-Rad Laboratory, Hercules, CA, USA). Protein quantification was analyzed using ImageJ software program (U.S. National Institutes of Health, Bethesda, MD, USA). The densitometry readings of the protein bands were normalized by comparing with the expression of β-actin.

### 2.10. Statistical Analysis

All the experimental results were expressed as the mean ± standard error of the mean (SEM) of triplicate samples. Significant differences between groups were calculated by one-way factorial analysis of variance (ANOVA) followed by a Bonferroni’s test and *p* < 0.05 was considered statistically significant. * *p* < 0.05 vs. Hep3B cell control, ** *p* < 0.01 vs. Hep3B cell control, *** *p* < 0.001 vs. Hep3B cell control.

## 3. Results

### 3.1. Scutellarein (SCU) Induces Cytotoxity in Hep3B Cells

We determined the cytotoxicity effects of scutellarein (SCU) in Hep3B cells by 3-(4,5-Dimethylthiazol-2-yl)-2,5-diphenyltetrazolium bromide (MTT) assay at various concentrations (0-, 100-, 200-, 300-, 400-, and 600-μM) for 24 h. As presented in Figure 1A, SCU exhibited a concentration-dependent inhibitory effect on Hep3B cells compared with the control at 24 h, and we estimated the 50% inhibitory concentration value as approximately 300 μM in the present study. We used SCU concentrations of 0-, 100-, 200-, and 300-μM for additional experiments. Along with the results obtained from MTT assay, microscopic examination revealed morphological changes including cell shrinkage, floating of dead cells, and decreased cell numbers in SCU-treated cells (Figure 1B), which further confirms the indications of apoptosis. SCU demonstrated a cytotoxic effect in human hepatoma Hep3B cells. The inhibitory effect of SCU is cancer-specific because it did not demonstrate any cytotoxicity in normal cells [14].

### 3.2. SCU Induces Cell Cycle Arrest and Regulation of Cdc25C, cdk1 and Cyclin B1 Protein Expression in Hep3B Cells

We investigated the effects of SCU on cell cycle arrest in Hep3B cells by flow cytometry analysis. The cells were treated with SCU at indicated concentrations (0-, 100-, 200-, and 300-μM) for 24 h and stained with propidium iodide (PI) followed by cell cycle distribution analysis. Figure 2A shows that the G2/M phase was significantly increased in SCU-treated cells compared with control cells. These data suggest that SCU induces G2/M phase arrest in Hep3B cells. To further analyze the effect of SCU on cell cycle arrest, we measured the expression levels of the proteins Cdc25C, cdk1 and Cyclin B1, which play a major role in cell proliferation and G2/M phase regulation. Immunoblot results revealed that cells treated with SCU showed reduced expression of Cdc25C, cdk1 and Cyclin B1 compared with the control in a concentration-dependent manner (Figure 2B).

### 3.3. SCU Induces Apoptosis in Hep3B Cells

We assessed the effect of SCU on inducing apoptosis in Hep3B cells was assessed by allophycocyanin (APC)/Annexin V and propidium iodide (PI) double-staining flow cytometry analysis. SCU treatment significantly increased early apoptosis, by 2.35%, 7.48%, 8.66%, and 23.1% at 0-, 100-, 200-, and 300-μM concentrations, respectively. Moreover it also significantly increased total apoptosis by 11.9%, 18.2%, 20.6%, and 30.2% at 0-, 100-, 200-, and 300-μM concentrations, respectively (Figure 3). In addition, the sub-G1 phase was significantly higher in the SCU-treated Hep3B cells (Figure 2A). These results clearly show that SCU induced early apoptotic cell death in Hep3B cells.

### 3.4. Effects of SCU on Nuclear Morphology in Hep3B Cells

We examined the effects of SCU on the morphological changes and damage to cell nuclei by the 4′,6-diamidino-2-phenylindole (DAPI) assay. The cells treated with SCU at indicated concentrations showed intense blue fluorescence representing chromatin pyknosis and also blue granules showing cleaved nuclei (Figure 4A). We performed a DNA fragmentation assay by 1.5% agarose gel electrophoresis on control and SCU-treated Hep3B cells, and observed a DNA ladder pattern in the SCU-treated Hep3B cells but not in the control cells (Figure 4B). These results suggest the formation of nuclear fragments due to apoptosis in SCU-treated Hep3B cells.

### 3.5. SCU Does Not Induce Apoptosis Through the Intrinsic Mitochondrial Apoptosis Pathway in Hep3B Cells

To confirm the apoptosis pathway, we examined the expression levels of mitochondrial apoptotic related proteins. We measured the pro-apoptotic protein Bax, anti-apoptotic protein Bcl-xL, and mitochondrial apoptotic pathway-related caspase-9 in SCU-treated Hep3B cells. Figure 5A shows that the Bax, Bcl-xL, and caspase-9 levels did not change in SCU-treated Hep3B cells. These results confirm that there is no involvement of the intrinsic mediated pathway in SCU-induced apoptosis in Hep3B cells. To confirm the changes in mitochondrial membrane potential (MMP), we measured the loss of MMP using the fluorescent dye 5,5′,6,6′-Tetrachloro-1,1′,3,3′-tetraethylbenzimidazolocarbocyanine iodide (JC-1) staining, and the results showed high MMP with intense red fluorescence, which indicates that the MMP did not change the cells (Figure 5B). These results further confirmed that SCU does not induce apoptosis via the intrinsic mitochondrial pathway in Hep3B cells.

### 3.6. SCU Induces Extrinsic Apoptosis in Hep3B Cells

To determine the involvement of the extrinsic apoptotic pathway in SCU-induced apoptosis in Hep3B cells, we measured Fas and Fas ligand (FasL) expression by immunoblotting. Western blot analysis revealed that SCU-treated cells up-regulated the expression in a dose-dependent manner. Death receptor 4 (DR4) expression was down-regulated (Figure 6). Similarly, among the downstream target proteins, Fas and Fas ligand (FasL) were unregulated, followed by down regulation of caspase-8 and caspase-3 and subsequent activation of cleaved caspase-8 and cleaved caspase-3, leading to increase of cleaved polymeric adenosine diphosphate ribose (PARP) expression. A substrate of caspase-3 was significantly activated in the SCU-treated group of Hep3B cells compared to untreated group cells (Figure 6). These results suggest that cell death in SCU-treated Hep3B cells occurs via extrinsic apoptotic pathway.

## 4. Discussion

Hepatocellular carcinoma (HCC) has increased threefold during the last 15 years and tops the mortality statistics for cancers (23.2%) [15]. More than 500,000 individuals suffer from this disease annually [16]. The preferred treatment for HCC is liver transplantation or resection, but despite surgical and regional therapies, prognosis remains unsatisfactory because of high tumor recurrence and tumor progression [17]. Recent interests in herbal therapeutic agents for treating diseases have gained attention due to their potential benefits and availability [18]. Flavonoids, as vital secondary metabolites, possess high-yield benefits in treating cancer with increased efficacy [19]. Growing evidence indicates that herbal extracts and their compounds, especially flavonoids, can inhibit cell growth and promote cancer cell death through different mechanisms including cell cycle arrest, autophagy, and apoptosis [20,21].

Compounds isolated from *Scutellaria baicalensis* exhibit pharmacological activities and have been recognized as valuable resources for the development of therapeutics. Scutellarein (SCU) is a major active flavonoid extracted from *Scutellaria baicalensis* with a wide range of biological activities such as anti-oxidant, anti-inflammatory and also anti-cancer effects by inducing apoptosis [22]. Accumulating research evidence has highlighted the potential role of (SCU) and its main metabolite SCU in the treatment of cancer [23].

In the present study, we examined the flavonoid SCU for its anti-cancer effects in hepatocellular carcinoma cells, Hep3B. Cell viability analysis revealed that SCU effectively inhibits cancer progression in Hep3B cell in a dose-dependent manner. We also investigated the mechanism by which SCU induces apoptosis in Hep3B cells.

Apoptosis plays an important role in developing and maintaining of tissue homeostasis. Intensive efforts have been made to explore the molecular mechanisms of the apoptotic signaling pathways from initiation to execution of apoptosis. Intensive efforts have been made to explore the molecular mechanisms of the apoptotic signaling pathways from initiation to execution of apoptosis [13]. The nuclear morphological analysis performed in this study also significantly depicted the induction of apoptosis [24]. DNA fragmentation assay revealed differently sized fragments of DNA in terms of apoptotic bodies, which confirmed that apoptotic cell death is induced by SCU in Hep3B cells. Similarly, flow cytometry analysis of SCU-treated Hep3B cells through allophycocyanin (APC)/Annexin V and propidium iodide (PI) double staining showed up-regulation of the apoptotic cell death fraction in the early stage, and the total apoptotic fraction also increased significantly.

In our study, SCU induced G2/M phase cell cycle arrest by inhibiting and the protein expression of Cdc25C, cdk1 and Cyclin B1. Previous studies showed that increases in the number of cells in the G2/M phase were clearly related to apoptosis [20]. Western blot analysis suggested that SCU induced apoptosis by activating caspases. The present study showed that treatment with SCU increased the expression of Fas ligand (FasL); FasL belongs to the tumor necrosis factor family and binds Fas, which leads to apoptosis in Fas-bearing cells [25]. The up-regulation of Fas ligand (FasL) can activate the apoptotic factors caspase-8 and caspase-3, leading to the cleavage of polymeric adenosine diphosphate ribose (PARP). In this study, Fas and Fas ligand (FasL) showed significant increases that activated cleaved caspase-8, caspase-3, and PARP, but the expression of the death receptor 4 (DR4) decreased. Thus, our results confirm that SCU induced Fas-mediated extrinsic apoptotic cell death in Hep3B cells.

In contrast, the mitochondrial-dependent apoptosis pathway, called the intrinsic pathway, was not activated in Hep3B cells treated with SCU. Overexpression of Bcl-xL, an anti-apoptotic protein, prevents cells from initiating apoptosis, while increase of Bax, a pro-apoptotic protein, induces the intrinsic apoptosis pathway [26]. In this study, we found that the levels of Bax and Bcl-xL and mitochondrial membrane potential does not change in SCU-treated Hep3B cells. In conclusion, SCU can inhibit the viability of hepatocellular carcinoma Hep3B cells via the Fas-mediated extrinsic apoptotic pathway. Findings in the current study strongly suggest that SCU can be used for developing potent anti-cancer agents for hepatocellular carcinoma Hep3B treatment.

## 5. Conclusions

The present study demonstrated the anti-cancer effect of scutellarein (SCU) in hepatocellular carcinoma (HCC) cells. The results obtained provided an understanding of its significant effect and also gave insight for developing SCU as a potent anti-cancer agent for HCC treatment. However, the present study has certain limitations with use of only one HCC cell line. Of note, upon consideration of the biological features of different HCC cell lines, replication of experiments is required to fully reveal apoptotic and cell cycle arrest pathways in HCC. Thus, further in vivo studies have been planned to consider future perspectives of research with respect to the in vitro analysis performed in the current study.

## Figures and Tables

**Figure 1 nutrients-11-00263-f001:**
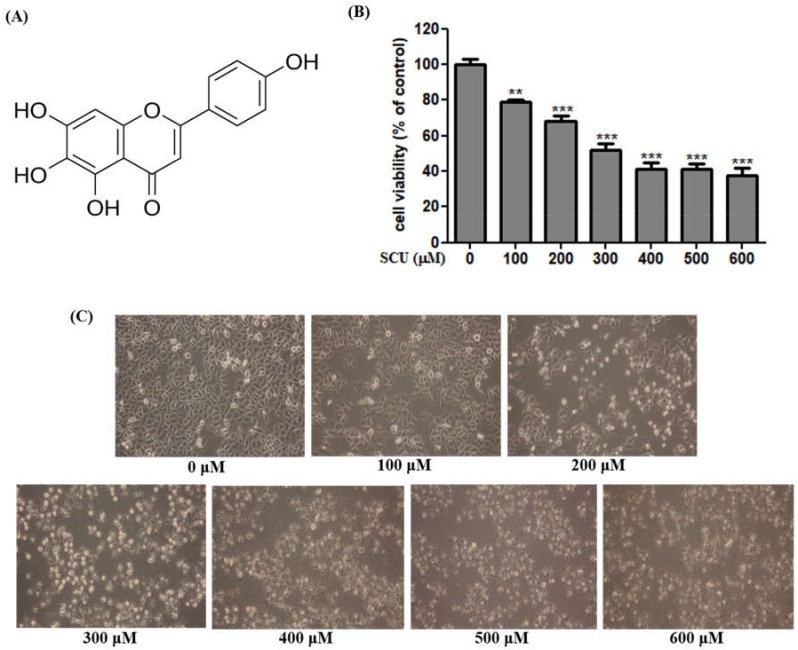
Scutellarein (SCU) inhibits proliferation of Hep3B cells. (**A**) Structure of SCU (**B**) The inhibition of cell viability was measured by 3-(4,5-dimethylthiazol-2-yl)-2,5-diphenyltetrazolium bromide (MTT) assay with various concentrations (0-, 100-, 200-, 300-, 400-, and 600-μM) of SCU for 24 h. Cell viability is represented in percentage relative absorbance compared to the controls. (**C**) Cellular morphology was observed under light microscope (magnification ×100). Values are given as the mean ± standard error of the mean (SEM). ** *p* < 0.01 vs. Hep3B cell control, *** *p* < 0.001 vs. Hep3B cell control.

**Figure 2 nutrients-11-00263-f002:**
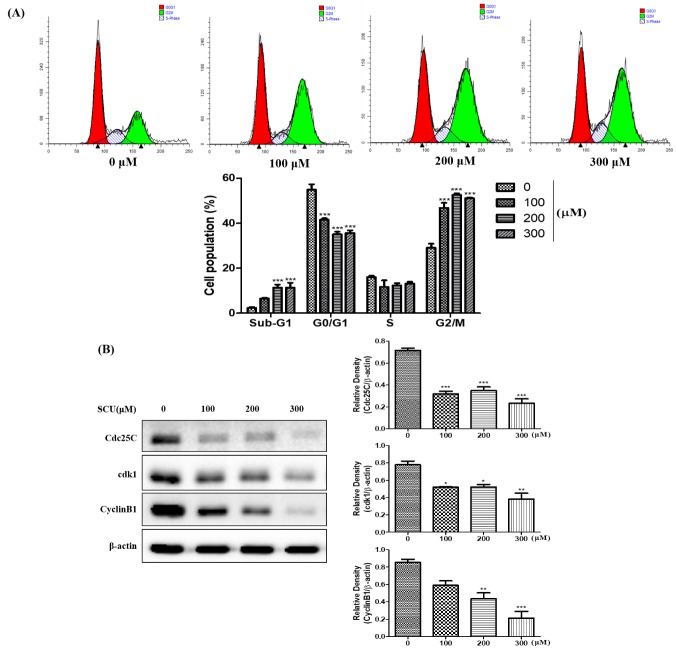
Scutellarein (SCU) induces G2/M cell cycle arrest of Hep3B cells. (**A**) To quantify the cell cycle phase distribution, the cells were treated with SCU at the indicated concentrations (0-, 100-, 200-, and 300-μM) for 24 h and stained with propidium iodide (PI) followed by flow cytometry analysis (red color: G0/G1 phase; white color: S phase; green color: G2/M phase). (**B**) The cells were treated with SCU at indicated concentrations (0-, 100-, 200-, and 300-μM) for 24 h. Total cell lysates were resolved in sodium dodecyl sulfate (SDS)-polyacrylamide gels and transferred onto polyvinylidene difluoride (PVDF) membranes. The membranes were probed with the Cdc25C, cdk1, Cyclin B1 antibodies. The proteins were visualized using electrochemiluminescence (ECL) detection system. Values are given as the mean ± standard error of the mean (SEM). * *p* < 0.05 vs. Hep3B cell control, ** *p* < 0.01 vs. Hep3B cell control, *** *p* < 0.001 vs. Hep3B cell control.

**Figure 3 nutrients-11-00263-f003:**
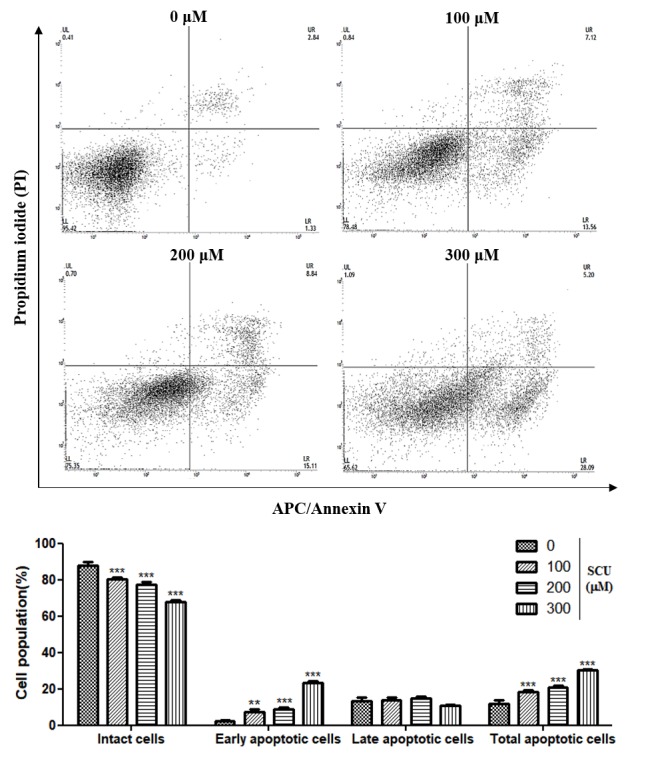
Scutellarein (SCU) induces apoptosis in Hep3B cells. To quantify the extent of SCU-induced apoptosis, the cells were untreated or treated with SCU at indicated concentrations (100-, 200-, and 300-μM) for 24 h. Allophycocyanin (APC)/Annexin V and propidium iodide (PI) double-staining was performed which was analyzed by flow cytometry. Values are given as the mean ± standard error of the mean (SEM). ** *p* < 0.01 vs. Hep3B cell control, *** *p* < 0.001 vs. Hep3B cell control.

**Figure 4 nutrients-11-00263-f004:**
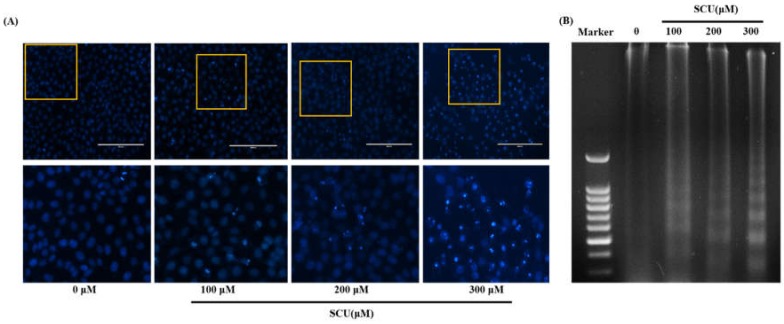
Nuclear morphological changes on scutellarein (SCU)-treated Hep3B cells. Hep3B cells were treated with SCU at indicated concentrations (100-, 200- and 300-μM) for 24 h. (**A**) After fixation, the cells were stained with 4′-6-diamidino-2-phenylindole (DAPI) to observe fragmented chromatin and apoptotic bodies. (**B**) DNA was isolated and electrophoresed in 1.5% agarose gel to confirm DNA fragmentation.

**Figure 5 nutrients-11-00263-f005:**
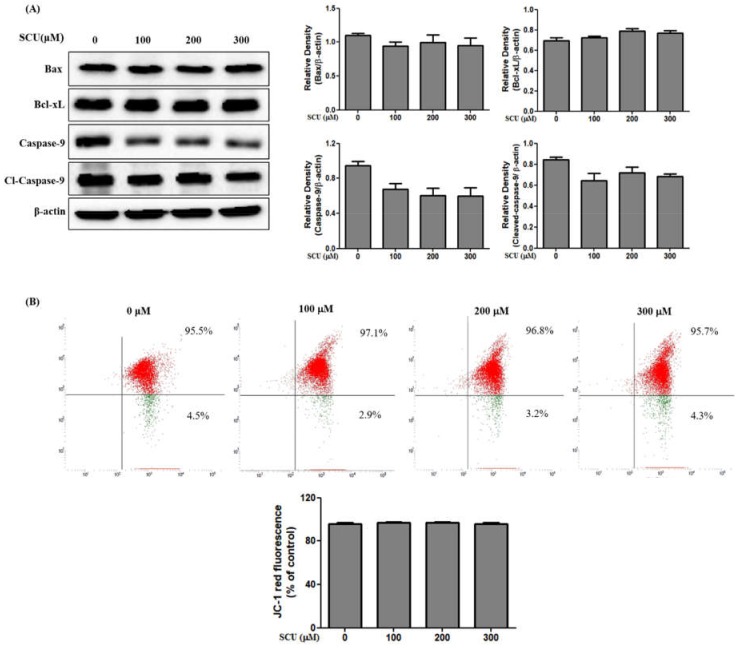
Scutellarein (SCU) does not induce the intrinsic apoptosis pathway in Hep3B cells. (**A**) The cells were treated with SCU at indicated concentrations (0-, 100-, 200-, and 300-μM) for 24 h. Total cell lysates were resolved in sodium dodecyl sulfate (SDS)-polyacrylamide gels and transferred onto polyvinylidene difluoride (PVDF) membranes. The membranes were probed with Bax, Bcl-xL, caspase-9, cleaved caspase-9, and β-actin antibodies. The proteins were visualized using electrochemiluminescence (ECL) detection system. (**B**) The cells were treated with SCU at indicated concentrations (0-, 100-, 200-, and 300-μM) for 24 h, washed twice with phosphate-buffered saline (PBS), and stained with 10 μM the fluorescent dye 5,5′,6,6′-Tetrachloro-1,1′,3,3′-tetraethylbenzimidazolocarbocyanine iodide (JC-1). The data shown are representative of three independent experiments with similar results.

**Figure 6 nutrients-11-00263-f006:**
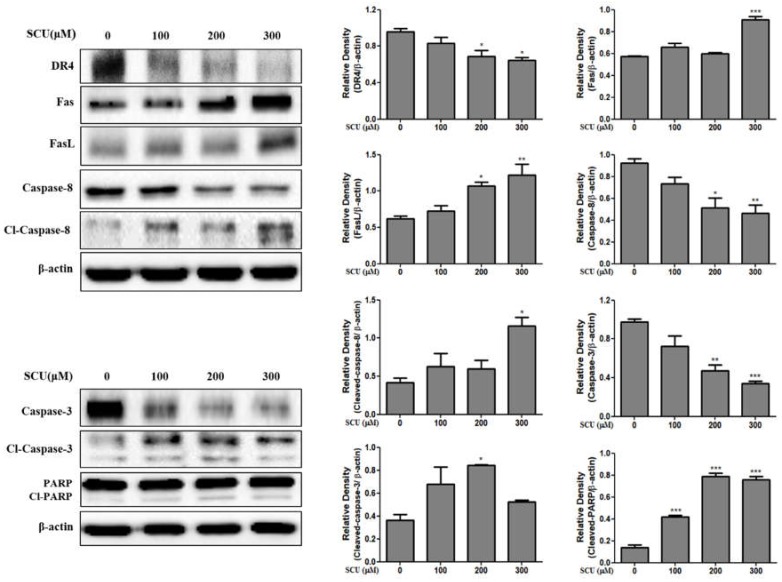
Scutellarein (SCU) induces Fas, FasL, caspase-8, and caspase-3 dependent apoptosis in Hep3B cells. The cells were treated with SCU at indicated concentrations (0-, 100-, 200-, and 300-μM) for 24 h. Total cell lysates were resolved in sodium dodecyl sulfate (SDS)-polyacrylamide gels and transferred onto polyvinylidene difluoride (PVDF) membranes. The membranes were probed with death receptor 4 (DR4), Fas, Fas ligand (FasL), cleaved caspase-8, cleaved caspase-3, cleaved- polymeric adenosine diphosphate ribose (PARP), and β-actin antibodies. The proteins were visualized using electrochemiluminescence (ECL) detection system. Values are given as the mean ± standard error of the mean (SEM). * *p* < 0.05 vs. Hep3B cell control, ** *p* < 0.01 vs. Hep3B cell control, *** *p* < 0.001 vs. Hep3B cell control.

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
