# Peer review of "Scutellarein Induces Fas-Mediated Extrinsic Apoptosis and G2/M Cell Cycle Arrest in Hep3B Hepatocellular Carcinoma Cells"

_nutrients, 2019, doi:10.3390/nu11020263_

Reviewer 1 Report

The manuscript describes the antiproliferative effect of Scutellarein on Hep3B cells. The author assert that the cell death induced by Scutellarein is due to the Fas mediated apoptotic effect on human hepatoma Hep3B cells. This mechanism of action could represent a new strategy for the chemotherapy of hepatocellular carcinoma.

 Major points to be revised

·         Although the study on cell cycle and on the expression of cell death markers performed by flow cytometry and western blotting suggest the antiproliferative effect of Scutellarein Fas mediated, mitochondria independent, in my opinion the authors should investigate in depth the modulation of mitochondrial membrane potential using flow cytometry to measure the loss of transmembrane potential not only measuring the JC-1 aggregates but also with TMRE staining to improve the data.

·         Moreover western blotting for cytochrome c is necessary to exclude its release from mitochondria to cytoplasm, contrary to Chui CH et al. 2005 and Chan JY et al. 2006.

·         Flow cytometry data appearing in figure 2 A show that the cell arrest in G2/M is more efficient when the cells are treated with 200μM respect to 300μM of Scutellarein. These data are in disagreement with the MTT, western blotting for cdc 25, cdk1 and CyclinB1and APC/Annexin V double staining results. In fact in these cases  it is clear that 300μM of Scutellarein reduced cell viability, blocked cell cycle and induced apoptosis, respectively, more than the lower concentrations. The authors should clarify this point better

·         More cell death markers must be added to evaluate a possible mechanism of necroptosis (MLKL, RIPK3, PS).

 Minor points

·         Materials and Methods: for western blotting analysis: it is necessary to indicate the dilutions of single antibody (1:1000 or 1:500) used and with which of these it is required milk or BSA to block the membranes. At line 156 there is one more point.

·         Results: At 3.2. section, line 199 there is a mistake TTo in place of To further

·         At 3.4 section it is not clear in figure 3 why statistical significant is appointed only on specific concentrations.  

·         In figure 6 it is clear that FasL increment is dose dependent but Fas increment is not dose dependent. Why Fas at 200 μM is reduced respect to 100 μM Scutellaria?

The signals of CI- Caspase 8 expression and CI caspase 3 are in disagreement respect to densitometric analysis.

·         Discussion: at line 290 the author mention at reference number 19 the apoptotic effects of Scutellaria. In this study on human gastric cancer AGS cells the effect of the molecule is mitochondrial dependent. So, it is necessary to explain why Scutellaria should act differently on apoptosis pathway, maybe due to different cell model ?

·         Reference: it is necessary to improve the discussion with the further references on Hep3B cells and Scutellaria, some of which are reported below.

1.      Chou CC, Pan SL, Teng CM, Guh JH. Pharmacological evaluation of several major ingredients of Chinese herbal medicines in human hepatoma Hep3B cells. Eur J Pharm Sci. 2003 Aug;19(5):403-12.

2.      Chui CH, Lau FY, Tang JC, Kan KL, Cheng GY, Wong RS, Kok SH, Lai PB, Ho R, Gambari R, Chan AS. Activities of fresh juice of Scutellaria barbata and warmed water extract of Radix Sophorae Tonkinensis on anti-proliferation and apoptosis of human cancer cell lines. Int J Mol Med. 2005 Aug;16(2):337-41.

3.      Chan JY, Tang PM, Hon PM, Au SW, Tsui SK, Waye MM, Kong SK, Mak TC, Fung KP. Pheophorbide a, a major antitumor component purified from Scutellaria barbata, induces apoptosis in human hepatocellular carcinoma cells. Planta Med. 2006 Jan;72(1):28-33.

4.      Tang PM, Chan JY, Au SW, Kong SK, Tsui SK, Waye MM, Mak TC, Fong WP, Fung KP. Pheophorbide a, an active compound isolated from Scutellaria barbata, possesses photodynamic activities by inducing apoptosis in human hepatocellular carcinoma. Cancer Biol Ther. 2006 Sep;5(9):1111-6. Epub 2006 Sep 28.

5.      Woo SU, Jang HR, Chin YW, Yim H. 7-O-Methylwogonin from Scutellaria baicalensis Disturbs Mitotic Progression by Inhibiting Plk1 Activity in Hep3B Cells. Planta Med. 2018 Sep 10. doi: 10.1055/a-0731-0394.

Author Response

Many thanks for reviewing our manuscript. We have changed the given comments in our manuscript. Because we need to include some references using Endnote program to avoid disorder of the reference sequence.

Reviewer 2 Report

Dear authors

Thank you for the opportunity to review your work.

There is no doubt that we are losing the fight against cancer. So any new strategy should looked as a new hope and taken very seriously.

I have a few questions/comments that I believe will improve the manuscript.

The finding that the level of DR4 being decreased can be reconciled with the activation of the apoptosis extrinsic pathway? The relevance/explanation of this finding should be discussed and clarified whether or not helping your conclusions.

The name of the compound should always be presented with small caps

cdks should be CDKs; cdc25C should be Cdc25C

What is the reason for the choice of the concentrations used?

The use of a student´s t test is not the most appropriate for the testing of means differences; use one-way ANOVA, followed by a post.hoc multiple comparisons test (Tukey's or Bonferroni's).

In the introduction (pg 2, line 55 and 56, the phrase: "Cell cycle arrest and induction of apoptosis are the important strategies in cancer therapeutics on the prevention of a wide variety of cancer" does not make sense. I believe it should be: "Cell cycle arrest and induction of apoptosis are important strategies in cancer therapeutic strategies."

some english corrections/improvements:

pg 1, line 41, remove therapeutic

pg 2, line 59, substitute cycline by cyclin

pg 3, line 110, insert a space after 12

pg 3, line 125, substitute was by were

pg 3, line 133, insert a space after 60

pg 3, line 133, substitute after treatment by and then incubated

pg 5, figure 1 legend: include statistics

pg 5, line 199, remove T before To

pg 5, line 203: include, in a concentration-dependent manner after control

pg 6, line 217: substitute and also signficantly increased the by  ,respectively. Moreover it also significantly increased

pg 6, figure 2 legend: include statistics

pg 7, figure 3 legend: include statistics

pg 7, line 228, include the before DAPI

pg 8, figure 5 legend: include statistics

pg 9, figure 6 legend: include statistics

pg 9, line 299 & 300, remove sentence that is repeated (Intensive...)

pg 10, line 306,substitute and the by . The

pg 10, line 308, include cells in the after number of

pg10, line 323, substitute Our findings by Findings

Author Response

(The authors gave the same response as above.)
